# Position: Trustworthy Model Context Protocol Enables Responsible Agentic AI!

**Arjhun Swaminathan** [* 1 2]  **Anika Hannemann** [* 3 4]

## Abstract

The Model Context Protocol (MCP) standardises AI agent-tool interaction, accelerating agentic AI adoption through interoperability. This presents an opportunity to embed trustworthiness. As a standard and an interface between agents and tools, MCP becomes a natural enforcement point; any improvements to it automatically propagate to all systems using it. Analyzing MCP through the European Commission's Ethics Guidelines for Trustworthy AI, we identify three things: fundamental shifts in how trustworthiness works, critical challenges these shifts create, and strategic intervention points where protocol-level mechanisms can achieve ecosystem-wide impact. We argue that MCP's architecture provides a foundation for trustworthiness and propose practical improvements to strengthen it. This position paper posits that building trustworthy MCP enables responsible agentic AI deployments.

The Model Context Protocol (MCP) is an open-source solution to standardise sharing context with language models, enabling the integration of tools into AI systems and supporting the development of composable workflows (Anthropic, 2024a;b). MCP follows a host-client-server architecture (cf. Figure 1) that makes it a central interface between language models and external tools, resulting in powerful AI applications. To illustrate, a clinical support system could retrieve both patient records from hospital databases and relevant medical literature, link and analyse them, and consult clinicians with recommendations for patient care.

These powerful applications require robust trustworthy AI guidelines. In practice, trustworthy-AI measures are often implemented at the application and organizational layers, which can lead to inconsistent adoption. MCP provides an opportunity to embed those guidelines at the protocol-level for two reasons. First, as MCP is becoming a widely adopted standard, incorporating trustworthy AI guidelines early helps ensure ethical considerations are built into the foundation of agentic AI. Second, the protocol's intermediary role within AI workflows provides a natural enforcement point, as any embedded improvements can propagate to all connected systems. Protocol-level implementations shift trustworthiness from being optional features to structural requirements. The evolution from HTTP to HTTPS illustrates this idea: security transitioned from application-level implementation to a protocol-level requirement, recognizing that inconsistent adoption creates systemic vulnerabilities (Kent, Stephen and Seo, Karen, 1999, p. 4, Kent, Stephen and Seo, Karen, 2005, p. 5).

This paper adopts the European Commission's Ethics Guidelines for Trustworthy AI (Commission, 2020) since they provide a concrete framework and complement the EU AI Act (European Parliament, 2024), which is the most comprehensive cross-sector framework at the time of this work. However, the principles it defines, including transparency, robustness, data governance, and accountability, are shared across major trustworthy-AI guidelines such as the OECD AI Principles (OECD, 2023), UNESCO's Recommendations on the Ethics of Artificial Intelligence (Zhu et al., 2025), and NIST's AI Risk Management Framework (NIST, 2024); the MCP-level implications outlined in this paper therefore apply broadly beyond the EU context. The principles include: human agency and oversight, technical robustness and safety, privacy and data governance, transparency, diversity, non-discrimination and fairness, societal and environmental well-being, and accountability.

For each principle, we identify limitations within MCP and its surrounding ecosystem and propose corresponding improvements, while considering the necessity of maintaining the protocol's lightweight nature. These improvements are classified into five categories, marked by [i]–[v] (cf. Call to Action 10), derived through iterative thematic clustering of the solution proposals identified in the analysis. The categories distinguish proposals according to the problem

---

[*]Equal contribution  [1]Medical Data Privacy and Privacy-preserving Machine Learning (MDPPML), University of Tübingen, Tübingen, Germany.  [2]Institute for Bioinformatics and Medical Informatics (IBMI), University of Tübingen, Tübingen, Germany.  [3]Swiss Centre for Responsible AI, Zurich, Switzerland.  [4]School of Engineering, Zurich University of Applied Sciences, Zurich, Switzerland. Correspondence to: Arjhun Swaminathan <arjhun.swaminathan@uni-tuebingen.de>, Anika Hannemann <anika.hannemann@scrai.ch>.

*Proceedings of the $43^{rd}$ International Conference on Machine Learning*, Seoul, South Korea. PMLR 306, 2026. Copyright 2026 by the author(s).

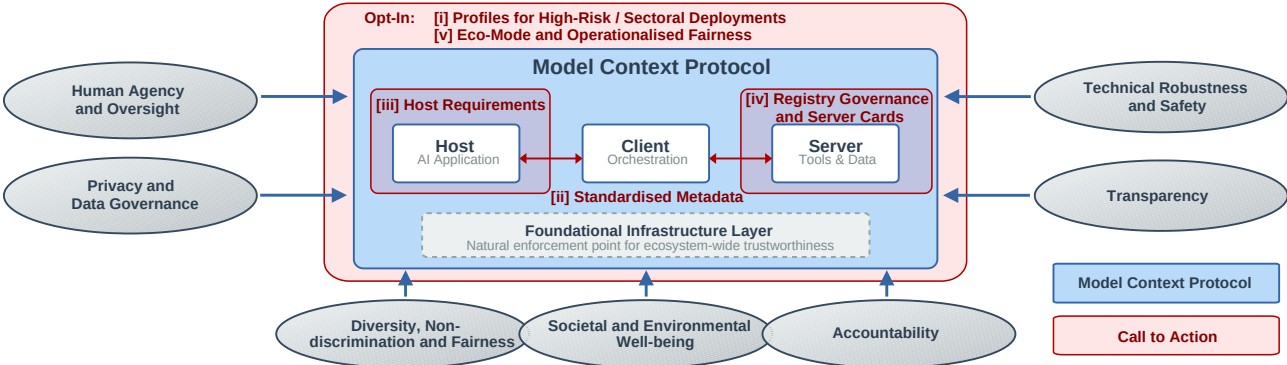

*Figure 1.* MCP architecture with our call to action to address the European Commission's Ethics Guidelines for Trustworthy AI.

they address and the point in the MCP pipeline at which they intervene:

**[i]** Risk-adaptive requirements keep MCP minimal but modulate safeguard triggers based on risk, domain, or deployment context.

**[ii]** Metadata standardisation introduces machine-readable information that persists across compositional workflows, enabling downstream enforcement of consent, provenance, and accountability.

**[iii]** Host-layer enforcement leverages the host's global view of the workflow to constrain execution through least-privilege scoping, audit logging, and risk-based approval.

**[iv]** Registry governance treats registries as surfaces that shape ecosystem-level trust through vetting, identity verification, and transparent ranking.

**[v]** Operationalised sustainability and fairness embed environmental impact, accessibility, and fairness into routine orchestration decisions as auditable constraints. Together, these categories map each solution to a specific intervention point in the MCP pipeline.

Certain challenges, however, remain unresolved and suggest directions for future research. We argue that MCP's architecture provides a foundation for trustworthiness and present a call to action: **Trustworthy Model Context Protocol Enables Responsible Agentic AI!**

## 0. Background

The host-client-server architecture determines where trustworthiness interventions can occur in MCP-mediated workflows. The *host* is the AI application that initiates and orchestrates the workflow. The *client* is the protocol component that maintains a connection to an MCP server and mediates requests and responses. The *server* exposes capabilities, such as tools, resources, or prompts, that can be invoked during the workflow. A typical MCP-mediated workflow therefore consists of a user goal being interpreted by the host, translated into one or more protocol-level requests

through the client, executed by one or more servers, and returned to the host as context for further model reasoning or user-facing output.

This decomposition matters for trustworthiness because it separates responsibilities across layers. Some requirements concern server behaviour, such as declaring tool capabilities. Others concern the host, such as logging. Registry-level mechanisms influence which servers are discovered, how they are presented, and which trust signals are available. We therefore use *trustworthy MCP* to refer to protocol-level and ecosystem-level mechanisms that support trustworthy AI requirements across MCP-mediated workflows.

The Ethics Guidelines for Trustworthy AI provide a normative lens for this analysis. We do not treat them as an MCP-specific checklist, but as a structured set of requirements for assessing whether MCP-mediated workflows preserve or weaken those principles. This framing allows us to ask which requirements can be supported by the protocol itself, which require host- or registry-level enforcement, and which remain open research or governance challenges.

## 1. Human Agency and Oversight

MCP reduces human-in-the-loop oversight by design. The EU Commission's guidelines do not require continuous manual intervention; they require *meaningful* oversight, defined as the ability to understand, intervene in, and correct AI behaviour (Commission, 2020). Risk-based and context-dependent mechanisms can satisfy this requirement without constant human involvement (Green, 2022, pp. 6–7, Santoni de Sio & van den Hoven, 2018, pp. 2–12). Reduced human-in-the-loop oversight therefore does not constitute a violation provided that appropriate mechanisms preserve alignment between system behaviour and human intent. The central concern is whether such mechanisms are in place.

**MCP and the challenge of informed oversight.** A core mechanism of MCP is explicit user consent for all data

access and operations (Anthropic, 2024a). By delegating the semantics and granularity of consent entirely to host implementations, MCP allows substantial variation in how much informed oversight users receive across compliant deployments. This variability is compounded by persistent informational asymmetries: requests are expressed in protocol-level terminology, offering limited understanding of agent intent, alternatives, or downstream consequences (Liao et al., 2020, p. 12). As a result, systems may function correctly under the protocol while remaining misaligned with user or policy expectations. Standardizing machine-readable disclosures of intent, scope, and impact would reduce this variability, giving hosts a consistent foundation for rendering human-readable action summaries **[iii]**.

**Human intent can degrade across translation steps.** Human intent traverses multiple steps: natural language to agent interpretation, agent reasoning to MCP requests, protocol formatting to external execution (Qu et al., 2025, pp. 5–6, Ji et al., 2023, p. 9). At each transition, fidelity to original intent weakens due to contextual noise or task drift (Laban et al., 2025, pp. 9–12), which could lead to context pollution. Further, bidirectional communication enables rapid request-response cycles without human intervention (Model Context Protocol, 2024). These iterations compound degradation at each translation step, producing value misalignment where outcomes diverge from original user intent (Ji et al., 2023, pp. 6–9). Mitigating this requires continuous host-side monitoring of context and task drift, with intervention triggered once predefined thresholds are exceeded **[i]**. MCP's elicitation primitive (Anthropic, 2025a) allows users to reinforce their intent, and the app architecture that enables tools to return interactive UI components directly in the conversation further strengthens oversight (Anthropic, 2026) **[i], [iii]**. Additionally, optional server-side risk metadata (tool risk categories, required approval levels) enables hosts to apply consistent escalation rules **[ii]**.

**Lifecycle phases create distinct oversight challenges.** During deployment, organizations lack standardised risk frameworks for assessing MCP servers (Campos et al., 2025, p. 3). During operation, interaction velocity overwhelms monitoring and makes consent prompts too frequent for review. During maintenance, audits are reactive; problems identified after execution. Automated audits catch technical violations but miss value misalignments and human audits face volumes exceeding practical capacity (Manheim & Homewood, 2025, p. 5). We propose adjusting oversight based on risk: requiring human approval for high-risk actions while allowing routine tasks to continue efficiently **[i], [iii]**. By combining threshold-based intervention, risk metadata, and human-readable disclosures, systems can maintain effective human control throughout the lifecycle.

## 2. Technical Robustness and Safety

AI systems must minimise harm (European Commission, 2019) across three dimensions: *robustness* addresses system behaviour under unexpected or changing conditions, *security* concerns adversarial attacks and malicious exploitation, while *safety* focuses on preventing unintended harm from design flaws and non-malicious interactions. MCP's distributed architecture and standardised interfaces introduce both traditional and novel challenges across all three dimensions. Current research measures attack resistance (Guo et al., 2025, pp. 3–9, Hou et al., 2025, pp. 14–28, Hasan et al., 2025, pp. 18–22). As robustness and safety are well-explored areas, we do not re-survey known attack vectors; instead, we focus on operational resilience and identify structural gaps that existing work does not address.

**MCP deployments are exposed to well-known security threats.** During the *creation* and *deployment* stages, developers face familiar attack vectors now manifesting in MCP's standardised interface design. When defining metadata and declaring capabilities, namespace typosquatting (Hou et al., 2025, pp. 15–16), tool poisoning (Narajala & Habler, 2025, pp. 4–5), and rug pulls (Hou et al., 2025, pp. 19–20) enable privilege escalation and supply chain compromise. During the *operation* stage, users encounter runtime threats including indirect prompt injection, command injection, and unauthorised access through unverified servers that can trigger data leakage or unauthorised actions (Hasan et al., 2025, p. 5). During the *deployment* and *maintenance* stages, ecosystem maintainers confront decentralisation-induced vulnerabilities where community-driven updates vary in quality and timing, causing inconsistent patch management and version control. This fragmentation increases risks of vulnerable versions, privilege persistence, and configuration drift.

**Registry-level trust does not guarantee runtime safety.** A key reason these threats persist is the architectural gap between registry-level trust and runtime execution guarantees. Registries validate namespace ownership and schema correctness but lack cryptographic or package-level validation mechanisms (Hou et al., 2025, p. 17, 29). Empirical analysis of 1,899 servers reveals 7.2% exhibit security vulnerabilities, yet traditional detection tools identify only a subset of MCP-specific risks (Hasan et al., 2025, p. 19). Although hosts must obtain explicit user consent before tool invocation, MCP does not enforce these security principles at the protocol level (Anthropic, 2025a). This leaves implementation quality as the sole defence mechanism, meaning even properly registered servers can execute arbitrary code with direct system access (Li & Gao, 2025, p. 10–12). Unlike PyPI's namespace verification challenges (Ruohonen et al., 2021, pp. 5–8), this gap is amplified because servers bridge

LLMs with databases, file systems, and APIs. The protocol validates intent through metadata but cannot guarantee alignment, creating a trust model where registry approval does not ensure execution safety.

**MCP's external touchpoints (dependencies, hosting, heterogeneous tools) undermine operational resilience.** MCP's external dependencies introduce vulnerabilities traditional frameworks cannot address. At the code level, empirical analysis reveals 66% of servers contain code smells, 14.4% have critical bugs, and 46% of Python package dependencies carry vulnerabilities (Hasan et al., 2025, p. 19, Ruohonen et al., 2021, p. 5). This is critical, since a single vulnerability in an open-source project can have far-reaching consequences and can potentially disrupt the global software industry, as the Heartbleed bug in OpenSSL showed (Walden, 2020, pp. 409–418). At the infrastructure level, remote hosting platforms like Cloudflare create cascading outage or configuration drift risks (Hou et al., 2025, pp. 27–28), while heterogeneous tool security models could cause format incompatibilities and information loss during internal communication (Hou et al., 2025, p. 27). Additionally, fault tolerance remains undefined; no standards govern agent behaviour when servers lag or fail. Critical scenarios (agent dropping, network outages, resource reallocation, workflow changes) lack resilience protocols (Hou et al., 2025, pp. 29–30). This is compounded by insufficient logging that prevents reproducibility and undermines post-deployment monitoring (Narajala & Habler, 2025, p. 2), leading to EU AI Act compliance gaps (European Parliament, 2024).

**MCP's internal complexities (compositional chains, distributed responsibilities) undermine robustness.** MCP fundamentally reframes robustness as a property of composition rather than individual component quality: Decision chains accumulate small errors, causing accuracy degradation only at workflow level, not in isolated tool testing (Laban et al., 2025, pp. 8–12). No certification frameworks exist to validate whether specific tool combinations behave safely, which pairings should be prohibited, or how compositional chains affect system reliability. This modularity creates trustworthiness fragmentation across four tiers (user, orchestration, agent, tool) without clear mapping to EU AI Act's provider/deployer categories. The question remains: who guarantees robustness at the workflow level when multiple independent actors contribute components?

**Component isolation enables safety engineering.** MCP's design includes features that support safety controls. Tools must be explicitly named, preventing agents from invoking random tools and creating clear behavioural boundaries (Anthropic, 2024a). Separating components into host, client, and server roles allows different security levels: servers can enforce strict access rules while individual tools

run in isolated sandboxes, containing potential failures (Narajala & Habler, 2025, p. 5). The protocol includes monitoring and logging capabilities that can meet EU AI Act post-deployment requirements (European Parliament, 2024). Operation and maintenance phases assume continuous monitoring (Hou et al., 2025, p. 9), providing a foundation for safety engineering, but these advantages only materialise through careful implementation.

## 3. Privacy and Data Governance

Privacy and data governance are among the most complex lenses to view MCP through, because the two most relevant regulations, the EU AI Act and the General Data Protection Regulation (GDPR), define different sets of actors and responsibilities. The EU AI Act focuses primarily on provider/deployer obligations, while the GDPR focuses on controller/processor roles; MCP instead specifies technical roles (host, client, server), and mapping these to legal roles is non-trivial in realistic multi-party workflows (Fischer, 2024, pp. 6–9,14–16). However, this decomposition can also be an upside: MCP-style architectures can make responsibilities more explicit when permissions, provenance, and accountability signals are attached to each component and data flow.

**Data minimisation is an engineering constraint.** Agentic AI workflows can undermine data minimisation by distributing and reusing context across tools and steps, often via automatic context assembly rather than manual curation. This results in (a) cross-tool exposure, where data revealed to one tool can be inferred or recombined with other tool outputs (Croce & South, 2025, p. 5), and (b) prompt-injection driven risks, where sensitive observations made during execution are induced to surface via tool calls (Alizadeh et al., 2025, p. 9, Hou et al., 2025, pp. 22–23). Minimisation must therefore be enforced throughout the workflow lifecycle, not only at the initial prompt boundary. A commonly recommended mitigation is least-privilege authorisation (including in official MCP documentation): scoping access via least-privilege authorisation scopes to what is necessary, reducing downstream leakage channels **[iii]** (Zhu et al., 2025, p. 11, Ntousakis et al., 2025, pp. 52–53).

**Purpose metadata across tool compositions.** Purpose limitation is difficult when data is reused beyond its original intent: once data enters shared context in MCP workflows, it may be repurposed for later tasks, forwarded to other tools, or transformed into summaries and derived artifacts that persist. A controller may collect data under specific terms, yet a tool could apply incompatible processing. Effective purpose limitation thus depends on specifying purposes concretely enough to be enforceable in system design and deployment (Von Grafenstein et al., 2022, pp. 8, 18). *Sticky*

*policies* (Pearson & Casassa-Mont, 2011, pp. 60-67), where machine-readable obligations travel with data across organizational boundaries, offer a template for preserving purpose across toolchains: attach purpose and usage constraints as metadata and validate them at each tool invocation rather than assuming initial consent remains applicable after transformations **[ii], [iii]**.

**Lawfulness, fairness and transparency require traceable data lineage.** Ensuring lawfulness, fairness and transparency is difficult in MCP workflows when multiple parties contribute or use data in shared context: MCP neither standardises nor mandates provenance, retention rules, or legal/governance roles and data-handling declarations, complicating production of transparency records for users and auditors. Building on provenance and lineage methods for ML pipelines (Schlegel & Sattler, 2025, pp. 4–9, Glavic & Dittrich, 2007, pp. 11–16), we recommend standardizing optional provenance metadata (e.g., source identifiers, timestamps, transformation steps) so systems can explain what happened to data across a toolchain **[ii]**. MCP already standardises an optional `lastModified` annotation, but richer provenance is neither standardised nor required; this gap also appears in ongoing discussions on the official MCP GitHub repository proposing standardised request/response annotations for trust and sensitivity. Governance metadata (declared roles, stated purpose, jurisdictional handling) could likewise be carried as anonymised metadata to preserve transparency across multi-party chains **[ii]**.

**Accuracy in MCP workflows depends on versioned provenance, not model correctness alone.** Accuracy in MCP workflows is tied to provenance and versioning: when context is built from retrieved documents, database queries, and tool-generated summaries, it may be unclear which source versions were used, when they were accessed, or whether later steps relied on incorrect data. Richer provenance metadata can mitigate this by letting hosts assess recency and trace how outputs were derived **[ii]**.

**Storage limitation must be implemented via lifecycle controls.** Storage limitation requires retaining data only for as long as necessary, including intermediate artifacts. In typical MCP workflows, context can persist across steps, and tool outputs and intermediate summaries can accumulate without explicit retention intent. Because MCP does not mandate retention rules or retention metadata (aside from task TTL metadata), hosts and servers must implement storage limitation via explicit policies and lifecycle controls rather than protocol defaults, consistent with the general security guidance in official MCP documentation. Provenance and lineage metadata can support this by identifying which artifacts exist, where they came from, and when they were created or accessed **[ii]**; together with retention metadata,

this enables fine-grained retention controls rather than a coarse global deletion rule, making storage limitation more systematic **[ii], [iii]**.

**Accountability requires mapping technical tool roles to legal responsibility.** Accountability is difficult in multi-party systems because responsibility must be allocated across actors who may be distributed, operate under different jurisdictions, and store logs or data in separate infrastructures. GDPR role assignment is already hard in cloud contexts and complex service chains (Fischer, 2024, pp. 6–9, 14–16), requiring careful analysis of who determines purposes and means at each step; MCP adds complexity because the host orchestrates tool calls while tools may independently process, store, transmit, or retain data. Improving accountability requires obligations and responsibility signals to remain attachable to specific processing steps, and demonstrability via compliance-relevant records like structured logging and documentation while avoiding logging that itself exposes personal data (Aghili et al., 2025, pp. 1–6). Provenance and lineage metadata can help auditors reconstruct which tools contributed which artifacts and how information flowed **[ii]**; governance metadata can further support responsibility allocation **[ii]**.

**Tool composition expands attack surfaces and governance gaps.** Integrity and confidentiality risks in MCP deployments depend both on implementation choices and on ecosystem governance. Even where authorisation is used, tool-augmented workflows can embed trust assumptions about connected servers and their behaviour once access is granted. Without strict least-privilege scoping and host-side isolation, a single compromised server can become a conduit for several privacy violations. Least-privilege authorisation for tool-calling agents reduces attack surfaces (Zhu et al., 2025, p. 11) and aligns with the official MCP recommended least-privilege scope model **[iii]**. Workflow-level security analyses further show that composition creates attack surfaces that do not appear when tools are evaluated in isolation (Ntousakis et al., 2025, p. 52, Croce & South, 2025, p. 5). Further, attacks such as prompt injection, tool poisoning, and compromised or impersonating servers that induce unintended disclosure via tool use can jeopardise confidentiality (Hou et al., 2025, pp. 15, 17–18, 20, 22–23). These risks are amplified by discovery and distribution dynamics. MCP does not currently define a protocol-level certification or trust-scoring scheme, so trust decisions are largely implemented by hosts and by registry or marketplace governance. While openness can broaden participation, assurance becomes uncertain unless registries provide robust review and responsive delisting **[iv]**. The official registry model relies on community flagging and delisting, which is helpful but not equivalent to systematic vetting. We therefore treat integrity and confidentiality as properties of the

broader tool and supply-chain ecosystem.

# 4. Transparency

Transparency can be framed as traceability of data and processes, explainability of decisions and deployment choices, and clear communication about system capabilities and limitations (European Commission, 2019). We discuss traceability and provenance-centric mechanisms for MCP workflows in section 3; here we focus on a complementary lever: standardised documentation artifacts that make MCP ecosystems interpretable to users, auditors, and deployers.

**Server Cards can operationalise transparency at registry level.** Building on established documentation templates (e.g., Model Cards and Datasheets) that standardise disclosures of intended use, limitations, and operational assumptions (Mitchell et al., 2019, pp. 222–223, Gebru et al., 2021, pp. 86–92), we suggest introducing registry enforced Server Cards as a listing requirement alongside `server.json`, extending emerging MCP discussions **[ii], [iv]** (MCP Blog, 2025). This disclosure would document, at minimum, publisher identity and contact, declared purpose, exposed tools, data access patterns and expected sensitive inputs/outputs, retention and sharing assumptions, security considerations, evaluation evidence where available, and maintenance policy, going beyond today's draft Server Card schemas, which primarily target discovery and configuration.

# 5. Diversity, Non-discrimination and Fairness

Trustworthy AI deployments must avoid unfair bias, ensure accessibility, and involve diverse stakeholders (Commission, 2020; European Parliament, 2024). MCP can threaten these through compositional bias propagation, missing metadata standards, and exclusionary registry governance.

**Composition forwards, amplifies, and introduces unfairness.** MCP's compositional workflows create bias amplification risks: protected attributes (race, gender, disability status) may be forwarded through workflows to unchecked tools that embed arbitrary biases without detection or intervention (Coppolillo et al., 2025, p. 11, Ferrara, 2024, pp. 2–4). Even with instructions to work against biases, Coppolillo et al. (2025, pp. 5–6) showed that agent discussions produced biased outcomes, creating echo-chamber dynamics that reinforced rather than challenged initial positions. Context generalisation without explicit fairness rules could default to behaviours that systematically disadvantage certain groups - for instance, assuming visual presentations work universally when they may not serve users with visual impairments (Guo et al., 2020, p. 4). Tool selection algorithms themselves may introduce bias in determining which tools are invoked for which users (Biswas & Rajan,

2021, pp. 983–989), and once sensitive attributes enter the workflow, no protocol-level mechanisms prevent their use by downstream tools lacking fairness safeguards.

**MCP would benefit from accessibility and fairness metadata.** Currently, the MCP protocol delegates accessibility and fairness responsibilities entirely to individual developers without providing standards or enforcement mechanisms, typically resulting in non-compliant tools (Anthropic). Adding structured accessibility metadata (alt-text, screen reader support) and fairness declarations (bias testing, protected attribute handling) (Mitchell et al., 2019, pp. 222–224) would enable systematic compliance with accessibility standards **[ii], [v]** (W3C Web Accessibility Initiative, 2024; W3C, 2017), help developers prioritise accommodations, and make bias testing and protected attribute handling auditable across the ecosystem. We propose adding fairness data to *Server Cards* **[ii], [iv], [v]**.

**Registry concentration creates walled garden dynamics.** Registry dominance by single companies, platforms, or jurisdictions creates walled gardens where discovery mechanisms favor certain perspectives while marginalizing others (Hou et al., 2025, pp. 10–11, Paterson, 2012, p. 100). Ranking algorithms may introduce bias that determines which tools are visible and adopted. Language barriers compound this issue: if MCP integration happens primarily in English, non-English-speaking communities face accessibility barriers for critical functions like data access requests (Orphanou et al., 2022). The question of who publishes MCP servers becomes a fairness concern: Evidence from Hugging Face, where 89% of developers remain isolated and a small number of companies control model adoption (Osborne et al., 2024, pp. 15, 17–18), suggests that MCP faces exclusion risks when participation barriers (technical expertise, resources, language) systematically disadvantage certain groups.

# 6. Societal and Environmental Well-being

Societal and environmental well-being extends the safety and fairness commitments beyond immediate users, spanning three dimensions: sustainable and environmentally friendly AI, social impact, and society and democracy (European Commission, 2019). MCP workflows and registries bring these concerns into focus because MCP lowers the cost of deploying tool-augmented agents and registries concentrate influence in discovery surfaces.

**Registries concentrate discovery power.** MCP positions itself as an open, platform-agnostic protocol, aligning with the EU's societal well-being objective of lowering barriers to access (European Commission, 2019). This is complemented by the official MCP Registry (Anthropic, 2025b),

which standardises discovery, while allowing sub-registries and client marketplaces to form. A common response to interoperability fragmentation is to pursue shared interface standards through multi-stakeholder standard-setting processes under neutral stewardship (Boehm & Eisape, 2021, pp. 3–4, 7). However, such standardisation rarely eliminates power. It can relocate it to discovery surfaces (such as registries, default configurations, and ranking systems) that can act as gatekeeping mechanisms (Brouwer, 2020, pp. 12, Lam, 2021, pp. 1–4, 44–46, Cowls et al., 2023, pp. 2–4). MCP stewardship moved to the Linux Foundation's Agentic AI Foundation (AAIF) while explicitly preserving its governance model. Official governance documentation (Anthropic) describes MCP as maintainer-led, with a defined, reviewable decision process. However, even with formal roles and procedures, foundations can still centralise influence by shaping ecosystem conventions and MCP's protocol-layer governance does not constrain how downstream hosts or third-party marketplaces implement ranking or defaults.

**Registry governance can function as a civic lever.** Societal well-being risks driven by power dynamics arise at the discovery surface. Server registries can effectively function as an agent marketplace by shaping visibility and providing trust signals. Research on app stores and platform-mediated search shows that ranking and selection mechanisms can create gatekeeping power (Brouwer, 2020, pp. 4-5, Lam, 2021, pp. 1–4, 44–46). Even with an open protocol, MCP may drift toward de facto concentration if discovery, ranking, and defaults are controlled by a small number of registries.

**We need a governance-by-design approach for registries.** Security risks also depend on governance of registries. Empirical analyses of MCP servers report security and maintainability issues at ecosystem scale (Hasan et al., 2025, pp. 19, 22, 27, 29, 32). Software supply-chain research shows that lookalike naming and semantic deception can induce misplaced trust (Neupane et al., 2023, pp. 3440–3447, 3452–3453). Similar failures may affect MCP tool discovery without mature package-management practices (Hou et al., 2025, pp. 15, 17–18, 29). We therefore motivate a governance-by-design approach for registries and client-side discovery interfaces **[iv]**. Concretely, this requires transparent and auditable ranking criteria, constraints on discriminatory defaults, and clear, contestable procedures for inclusion, de-listing, and appeal. These risks intensify further in the future if tool discovery couples to dominant intermediaries such as app stores and platform search, where ranking and defaulting can act as gatekeeping mechanisms (Brouwer, 2020, pp. 11–12, Lam, 2021, pp. 1–4, 44–46).

**Media integrity can be at risk in MCP workflows.** Disinformation risk in MCP-mediated news workflows has two complementary forms: LLMs can generate realistic news-style narratives at scale (Zellers et al., 2019, pp. 8–9), and agentic tool pipelines remain vulnerable to workflow manipulation (Hou et al., 2025, pp. 18, 25), where malicious instructions can be embedded in retrieved content or intermediate artifacts and then executed through tool use. Media integrity in MCP workflows should therefore be treated as a software supply-chain and governance problem (Hasan et al., 2025, pp. 10, 19, 29, Hou et al., 2025, pp. 15, 17–18, 29). Registries can amplify these risks because registries determine which news-facing tools are most visible and can attach trust signals that shape what is treated as reputable. Multi-agent debate or cross-checking can help, but adversarial behaviour in collaborative settings can still steer group outcomes toward incorrect conclusions (Amayuelas et al., 2024, pp. 6933–6936, Wynn et al., 2025, p. 10). Trustworthy MCP-mediated news workflows therefore require governance-by-design at the discovery and host surfaces: stronger publisher identity verification, conservative defaults for civic domains, and contestable processes for de-listing, appeal, and independent oversight **[i], [iii], [iv]**.

**Rapid adoption can intensify labour disruption.** MCP lowers the engineering and coordination cost of deploying tool-augmented agents in real workflows, accelerating adoption of agentic automation. The concern is not only whether work changes (Brynjolfsson & Mitchell, 2017, pp. 1531–1534), but how quickly and for whom (Frey & Osborne, 2017, pp. 39–48, Eloundou et al., 2023, pp. 10–20). Impacts are rarely all-or-nothing at the occupation level: automation can displace some tasks while complementing or enabling others (Brynjolfsson & Mitchell, 2017, pp. 1531–1534, Acemoglu & Restrepo, 2018, pp. 203–207). MCP acts as an accelerator whose effect depends on deployment speed relative to institutional adaptation and job creation; this is not a claim that MCP uniquely causes displacement, but that interoperability infrastructure can shift adoption rates and thereby intensify consequences if adaptation lags behind (Acemoglu & Restrepo, 2018, pp. 226–229, Kulveit et al., 2025, pp. 2–7, Mazeika et al., 2025, pp. 1–3, 11–12). To reduce autonomy loss under faster rollout, MCP deployments should include participatory rollout mechanisms (Nitsch et al., 2024, pp. 263, 265) at the organisational level that respect worker preferences about their involvement in agent workflows **[i]** (Shao et al., 2025, pp. 8–17).

**Workflow design determines environmental footprint.** MCP reduces integration overhead, but multi-step tool use can expand computation per task; energy usage depends on workload and orchestration, not only on model size (Fernandez et al., 2025, pp. 4–6). Longer-context operations add substantial memory and compute overhead, increasing workflow footprint (Yang et al., 2025, pp. 9, Choi et al., 2025, pp. 1–2). Repeated retrieval outputs and tool results can bloat context, and multi-step chains amplify the footprint.

**MCP as an interface to reduce environmental costs.**
Prior work argues that AI systems should report resource "price tags" and incorporate efficiency into evaluation (Schwartz et al., 2020, pp. 56, 59–60, Strubell et al., 2019, pp. 3648–3649, Patterson et al., 2021, pp. 15). MCP provides a natural point to measure and log energy consumption. We propose an optional *Eco Mode* profile for energy-aware orchestration that layers host and registry level conventions on existing MCP controls **[i], [iii], [v]**. Eco Mode could (a) set explicit per-request token caps via MCP Sampling's `maxTokens`, and include `modelPreferences` (e.g., higher `costPriority`) as non-binding hints to prefer cheaper models, (b) introduce explicit host enforced *tool-call budgets* (e.g., maximum calls per workflow and per tool) and operationalise host-side rate limiting by making limits explicit and auditable, and (c) apply restrictive default thresholds that reduce unnecessary tool use. Convenient synergies can happen: Budget-aware strategies show that agents achieve better cost-performance scaling when they reason under tool-call budgets (Liu et al., 2025, pp. 11–14), and alignment methods can reduce unnecessary tool invocation (Wang et al., 2025). Eco Mode can further promote cost-effective usage by having hosts route narrow tasks to smaller/cheaper models when appropriate (e.g., via sampling configuration and model-selection policy), since in some settings specialised small language models can outperform larger models **[iii], [v]** (Jhandi et al., 2025, pp. 5). To support measurement and accountability, MCP registries or downstream aggregators could optionally publish publisher-provided cost and energy metadata for servers, while hosts log per-workflow usage estimates to enable auditing **[iii], [iv], [v]**. These disclosures should be treated as claims rather than facts: Eco Mode does not itself solve environmental accountability, but creates a shared exposure surface on which claims can be compared, challenged, and governed. Hosts could benchmark declared metadata against observed runtime behaviour, such as token counts, tool-call volumes, latency, and budget-limit violations, while registries or downstream marketplaces could use persistent divergence as a signal for downranking, flagging, or delisting **[iii], [iv], [v]**. This raises the cost of inflated claims without requiring centralised enforcement, since discoverability and reputation become coupled to consistency between declared and observed behaviour. Designing optimal penalties, audit probabilities, and reward structures for such disclosures remains an open mechanism-design problem (Christoph, 2025). Finally, Eco Mode could standardise an optional energy-efficiency hint (e.g., carried as metadata) so servers can switch to energy-efficient variants when they exist **[ii], [v]**. Together, these measures preserve MCP's interoperability while better aligning agentic workflows with environmental well-being.

## 7. Accountability

Accountability requires that responsibility for an AI system and its outcomes can be established and upheld across the full lifecycle, and it is closely linked to fairness because it is what makes unjust adverse impacts contestable rather than merely observable (European Commission, 2019). This is operationalised through auditability, minimisation and reporting of negative impacts, explicit handling of trade-offs, and redress. MCP workflows strain these because responsibility is distributed across hosts, servers/tools, registries, and external data sources, and because toolchains can change over time. This can be an upside, since it delineates responsibility boundaries, but risks accountability gaps if obligations are not attached to each step in a verifiable way (Cobbe et al., 2023, pp. 1192–1195, Fischer, 2024, pp. 18–20, 24–26).

**Auditability requires monitoring across evolving toolchains.** Accountability must cover ecosystem dynamics, where third-party MCP servers can change behaviour (Hou et al., 2025, pp. 15, 19) over time or become compromised, resembling software supply-chain attacks (Ohm & Stuke, 2023, pp. 3–11, Ladisa et al., 2023, pp. 6–9). While MCP provides a logging mechanism that allows servers to emit structured log messages to clients, it does not mandate what must be logged, where logs are stored, or how long they are retained. This makes auditing fragile: overly sparse logs can prevent reconstruction, while overly detailed logs can create privacy risks (Jain et al., 2016, p. 14). Accountability requires treating hosts **[iii]** (and, when relevant, registry/marketplace operators **[iv]**) as enforcement points and for MCP to require a minimal, standardised audit record for high-risk deployments: covering tool identity and version, authorisation context, inputs/outputs at an appropriately interpretable level, and policy constraints, with privacy-by-design and lifecycle controls such as version pinning and provenance metadata.

**Standardizing redress in MCP.** In MCP workflows, redress requires (a) attribution signals that link an outcome to the specific host decision and the server/tool invocations that contributed to it, and (b) records that are sufficient for independent review without turning logs into a privacy risk. We therefore posit that MCP should evolve toward a minimal, interoperable redress interface standardizing (a) provenance metadata **[ii]**, and (b) a structured, host-generated redress summary (with a well-defined schema) that can be rendered to users and provided for audit, summarizing the toolchain at an appropriate abstraction level **[ii], [iii]**.

## 8. Cross-Effects Between Principles

The seven principles are analytically distinct but operationally intertwined. MCP's distributed architecture couples

them: a failure in one principle weakens others, and mitigations for one can heighten risk in another. Some principles encode competing goals. The examples below illustrate these types rather than comprehensively enumerate them.

**Failure propagation** occurs when context drift in human agency (Section 1) does not merely reduce oversight; it amplifies fairness failures when degraded intent systematically advantages certain user groups (Ferrara, 2024, pp. 1–2). Compromised tool integrity (technical robustness) becomes a privacy leak if that tool accesses sensitive data (Hou et al., 2025, pp. 18–19). Weak registry governance (transparency/accountability) leaves biased or vulnerable servers discoverable and trusted (Hou et al., 2025, p.15, 33). **Mitigation trade-offs** arise when safeguards for one principle heighten risk in another. Detailed audit logs strengthen accountability but require privacy-aware design to avoid recording sensitive attributes (Aghili et al., 2025, p.2). Purpose metadata (privacy) requires careful implementation; if it creates participation barriers, it can reduce diversity and entrench fairness problems. Least-privilege scoping (privacy/security) can limit which tools hosts access. **Normative conflicts** emerge where principles encode competing goals. Individual privacy conflicts with group fairness, which requires protected attributes for bias auditing (Kazim & Koshiyama, 2021, p.220, 223). Transparency (detailed disclosures) can harm individuals' privacy if it reveals decision proxies for sensitive attributes (Kazim & Koshiyama, 2021, p.221). Registry governance decisions (cf. section 6) propagate across all principles by shaping server discoverability, with consequences for security, fairness, and accountability.

## 9. Alternative Views

A central claim of this position paper is that MCP is a high-leverage interface for operationalising trustworthy AI principles in agentic systems. An alternative view is that responsible practices belong primarily at the application and deployment levels, enforced through organisational processes, sectoral regulation, and provider/deployer obligations. We agree that a protocol by itself cannot guarantee trustworthiness. Our narrower claim is that MCP can standardise shared technical hooks (e.g., provenance mechanisms and audit-relevant metadata) that make trustworthy deployment practices easier to implement consistently.

A second view is that adding responsibility mechanisms at the protocol layer increases complexity and resource overhead, turning MCP from a lightweight interoperability interface into a burden that developers and deployers may bypass. This concern is strongest when safeguards are rigid or universal. A complementary view is that requirements for safety, transparency, redress, and sustainability are context-dependent, so protocol-level mandates will either over-constrain low-risk deployments or under-protect high-risk ones. We agree on proportionality and on preserving a lightweight core. Our position is therefore that MCP should remain minimal by default, while providing optional profiles and standardised mechanisms that hosts, registries, and governance frameworks can require in higher-risk settings.

## 10. Call to Action

**[i]: Keep MCP minimal.** Maintain a lightweight core protocol, while defining opt-in measures that can be deployed in high-risk or sectoral deployments (e.g., assisting monitoring of task or context drift, assisting in elicitation of human approval for high-risk actions).

**[ii]: Standardise metadata.** Define interoperable metadata that persists across decision chains: richer provenance, per-tool-call identifiers, tool risk categories, purpose and usage constraints, and governance metadata.

**[iii]: Make hosts an enforcement point.** Publish reference host requirements for: least-privilege scoping, risk-adaptive human approval, structured privacy-aware audit trails, and a minimal redress interface.

**[iv]: Treat registries as governance surfaces.** Adopt governance-by-design for discovery: transparent ranking criteria, non-discriminatory defaults, and contestable inclusion/delisting/appeal. Require Server Cards for listing (identity/contact, declared purpose, tools, data access expectations, retention/sharing assumptions, update policy, trustworthiness data including fairness and accessibility characteristics).

**[v]: Operationalise sustainability and fairness.** Introduce Eco Mode (token/tool-call budgets, rate limits, energy/cost metadata) and standardise accessibility/fairness declarations so responsible practices become visible and auditable.

## 11. Conclusion

MCP is becoming a common interface for agentic AI deployments. This position paper argued that trustworthiness in such systems is increasingly a property of composition and ecosystem dynamics, not only individual models or tools. Viewed through the EU Ethics Guidelines for Trustworthy AI, MCP's standardisation creates both risk and opportunity: failures can propagate across deployments, but so can well-designed safeguards. Our central claim is therefore pragmatic: MCP should remain lightweight while standardizing responsible-AI measures, thereby making trustworthy agentic AI easier to build and harder to circumvent. We hope this work helps align MCP's technical improvements and ecosystem governance with trustworthy outcomes at scale.

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
