# OpenReview forum: "Position: Trustworthy Model Context Protocol Enables Responsible Agentic AI!"
_ICML.cc/2026/Position_Paper_Track — ICML 2026 Position Paper Track regular_

### Official Review · Reviewer_x9Nk · 2026-02-22

**Significance:** 3
**Argument Clarity:** 3
**Rating:** 4
**Confidence:** 2

**Questions:**

See weakness.

**Alternative Views Section:**

Yes

**Compliance With Llm Reviewing Policy A Conservative:**

Affirmed.

**Discussion Potential:**

3

**Final Justification:**

My concerns addressed. However, not fully clear of all trustworthy AI claims so have low confiidence.

**Paper Summary:**

This paper propose to build a trustworthy model context protocol. Based on EU Commission's ethics guidelines, the authors propose trustworthiness for MCPs including human agency and oversight, robustness and safety, privacy, transparency, diversity and fairness, well-being and accountability. The authors provide requirements for trustworthy MCPs and suggestions on current guidelines and actionable suggestions for trustworthy MCP.

**Position:**

Yes

**Position In Title:**

Yes

**Related Work:**

3

**Strengths And Weaknesses:**

Strength.

This paper summarize and describe how to build trustworthy MCPs based on regulation requirements of EU. The authors make deep surveys about algorithmic, technical and engineering aspects of MCPs, which could be highly important when building MCPs for real-world use.

Weakness.

1. While the overall focus on trustworthy MCPs is a plus, the paper covers almost every aspect of trustworthy AI, which makes the suggestions overly broad and lack specific focus. For example, in the specfic area of robustness and safety, the area contains hundreds of papers on attacking MCPs, yet contribute only one section (1/10) of the overall paper. As such, the overall paper looks like a comprehenesive policy summary rather than an actionable work,  since each of the section requires comprehensive effort.

2. There lack a formal definition of MCP and its trustworthiness. While I admit many surveys do not contain this, adding a rigorious math formulation would add clarity to the discussion.

**Support:**

3

---

> ### Author Rebuttal · Authors · 2026-03-30
>
> Dear Reviewer,
>
> thank you very much for your review. Please find our rebuttal here:
>
> >>Strength.
> >>This paper summarize and describe how to build trustworthy MCPs based on regulation requirements of EU. The authors make deep surveys about algorithmic, technical and engineering aspects of MCPs, which could be highly important when building MCPs for real-world use.
>
> Thanks for appreciating the depth of our surveys and the range of things covered. Below, we provide detailed responses to each of the raised concerns and hope these clarifications fully resolve them.
>
>
> >>Weakness.
> >>While the overall focus on trustworthy MCPs is a plus, the paper covers almost every aspect of trustworthy AI, which makes the suggestions overly broad and lack specific focus. For example, in the specfic area of robustness and safety, the area contains hundreds of papers on attacking MCPs, yet contribute only one section (1/10) of the overall paper. As such, the overall paper looks like a comprehenesive policy summary rather than an actionable work, since each of the section requires comprehensive effort.
>
> We appreciate the reviewer’s point. Robustness and safety are indeed highly developed areas with extensive existing surveys, which is why we only summarize key references and point to existing work rather than replicate or extend that work. We highlight under‑discussed yet important topics, such as the protocol’s environmental impact and registry governance, as they receive far less attention in the literature but are essential for building trustworthy MCP deployments in practice. Our intention is not to solve each area fully, but to map regulatory demands to engineering needs and highlight where guidance is still lacking - something policy summaries do not provide. But we agree with the reviewer that the objective of the paper can be stated more explicitly and we plan to do so in the revised version.
>
> >>There lack a formal definition of MCP and its trustworthiness. While I admit many surveys do not contain this, adding a rigorious math formulation would add clarity to the discussion.
>
> We agree that a more explicit definition of MCP would strengthen the paper. By the ICML submission rules an additional page for the final version is allowed. In the beginning of the manuscript, we plan to include a short section that formalizes MCPs, their actors, and context on the EU Commission's guideline.
>
> Given that MCPs are increasingly becoming the backbone of emerging agentic AI systems, we see this topic as both timely (or even urgent) and highly relevant. We will incorporate your suggestions into the final version and hope this and our clarifications address your concerns and support a reassessment of the paper’s contributions. We thank you for your constructive feedback and time!

---

> > ### Author Rebuttal · Reviewer_x9Nk · 2026-04-01
> >
> > After reading this I would say the paper and the rebuttal are quite good. I have raised my score but am not fully confident about my evaluation since I'm not working on many part of the topics mentioned.

---

### Official Review · Reviewer_aBB3 · 2026-03-11

**Significance:** 3
**Argument Clarity:** 1
**Rating:** 4
**Confidence:** 4

**Questions:**

1. How were the 5 categories [i]-[v] identified, and what purpose do they serve in supporting the paper’s stated position?
2. Are the key bottlenecks to improving the trustworthiness of MCP institutional or technical?

**Alternative Views Section:**

Yes

**Compliance With Llm Reviewing Policy A Conservative:**

Affirmed.

**Discussion Potential:**

3

**Final Justification:**

The author(s) have/had meaningfully interacted with my concerns, though given the review process, I am unable to assess the quality of revisions made - some of which will be substantial. As such, the paper is still borderline in my opinion, though shows promise.

**Paper Summary:**

This paper argues that MCP represents a promising opportunity for operationalising responsible AI practices across the AI agent ecosystem. The paper analyses MCP with respect to fulfilment of the European Commission’s Ethics Guidelines for Trustworthy AI, and suggests potential modifications to better enable responsible agentic AI practices.

**Position:**

Yes

**Position In Title:**

Yes

**Related Work:**

2

**Strengths And Weaknesses:**

This paper touches on a novel and important topic, and the position advocated for is timely. The paper’s methodology of drawing on a specific set of ethical guidelines and principles is sound, and provides a good anchor point for analysing MCP in its current state, as well as potential improvements that could help realise the ethical principles.

However, the paper suffers from presentational limitations that make it a challenging read and severely hinder its clarity and understandability.

For a start, there is very limited background context on both MCP and the EU ethical principles that form the basis for this work. I would suggest adding a short section after the introduction that provides a brief overview of these two central components to the paper, to ensure that all readers have the required context to interpret the rest of the paper.

Relatedly, the five categories [i]-[v] are not introduced before being used to ‘tag’ certain aspects of the discussion in sections 1-7. Definitions and descriptions of these categories, what they represent, and the purpose that they serve in the paper should appear clearly before they are referenced in the main body of the paper. (I appreciate the brief descriptions in Section 9, though I find that this still lacks context on how they were identified, and the purpose they serve in supporting the paper’s main argument.)

Additionally, it is not clear what the bold headings of each paragraph are, and there is limited consistency between them. For example, the majority seem to point to current limitations of MCP for realising an ethical principle. But others focus more on solutions (e.g. *“Server Cards can operationalize transparency at the registry level.”*) This may sound minor, but I found that it meaningfully contributed to the challenge of interpreting and navigating this paper’s content. Clearer organisation of each section, for example into ‘Current Limitations of MCP’ and ‘Potential Solutions’ subsections would go a long way here.

Finally, the text in Figure 1 is far too small to be read easily, and I find that Figure as a whole lacks sufficient explanation in the main text body or caption to be easily interpreted.

Overall, this paper could make a meaningful contribution to the discussion but is not fit for publication in its current state.

**Support:**

2

---

> ### Author Rebuttal · Authors · 2026-03-30
>
> Dear Reviewer,
>
> thank you very much for your review. Please find our (slightly condensed due to character limits) rebuttal here:
>
> >> For a start, there is very limited background context on both MCP and the EU ethical principles that form the basis for this work.  [...]
>
> We agree that a more explicit definition of MCP and the EU principles would strengthen the paper. By the ICML submission rules an additional page for the final version is allowed. We plan to include a short section that formalizes MCP, their actors, and context on the EU Commission's guideline. We also note that our paper is intended for a broad audience - not only technically focused readers but also those interested in governance, regulation, and responsible data practices. Since the EU is at the forefront of shaping responsible AI and data governance frameworks, adding this background will help situate the paper more clearly for all readers. Thank you for pointing this out.
>
> >> Relatedly, the five categories [i]-[v] are not introduced [...].
>
> We thank the reviewer for this suggestion. We acknowledge that, in conducting a deep dive into the topic, certain foundational elements may have been assumed rather than explicitly described. Although we briefly introduce categories [i]–[v] in the introduction (with a pointer to Section 9), we agree that they should be defined more clearly before they are used to tag the discussions in Sections 1–7. The additional page permitted in the final version will allow us to expand these descriptions and provide the necessary context.
>
> >> Additionally, it is not clear what the bold headings of each paragraph are [...]
>
> The bold headings in each paragraph are intended to highlight key claims or findings to guide the discussion across the broad set of topics we address. Further structuring would certainly be desirable; however, given the breadth of material and strict page limits, we prioritized content over adding additional subsections. Moreover, the principles we discuss are independent and interconnected, making a single uniform structure impractical and requiring some degree of tailoring for each section. In practice, we found that some principles have well‑defined limitations but no established solutions, while others are supported by emerging practices rather than well‑defined problem statements. This heterogeneity made a rigid, categorical structure difficult to apply consistently across all sections. That said, we agree that greater consistency in how these headings are introduced and used could improve readability. In the revised version, we will refine these headings and, where appropriate, clarify whether they refer to a limitation, an emerging solution, or a key observation. This should help guide the reader more clearly without forcing a rigid structure that does not fit all principles.
>
> >> Finally, the text in Figure 1 is far too small to be read easily, and I find that Figure as a whole lacks sufficient explanation in the main text body or caption to be easily interpreted.
>
> Thank you for pointing that out, we will increase both the figure and the text and provide a detailed caption.
>
> >>  How were the 5 categories [i]-[v] identified, and what purpose do they serve in supporting the paper’s stated position?
>
> The five categories [i]–[v] were identified by clustering the solution proposals identified in our analysis. Clustering them allowed us to distill a large set of heterogeneous solutions into a coherent structure. These categories serve two purposes: (1) they improve readability, and (2) they enable us to map each solution to the MCP architecture shown in the visualization, helping readers see where in the pipeline specific interventions apply. This makes the overall analysis more interpretable and directly actionable.
>
> >> Are the key bottlenecks to improving the trustworthiness of MCP institutional or technical?
>
> Both. Improving MCP trustworthiness requires institutional and technical progress. Institutions maintaining MCP servers must prioritize responsible practices and governance, and the broader community needs to actively support this. At the same time, research must advance technical methods without unnecessarily constraining MCP’s capabilities. Meaningful progress depends on coordinated efforts across both dimensions.
>
> >> Overall, this paper could make a meaningful contribution to the discussion but is not fit for publication in its current state.
>
> Given that MCPs are increasingly becoming the backbone of emerging agentic AI systems, we see this topic as both timely (or even urgent) and highly relevant. We will incorporate your suggestions into the final version and hope this and our clarifications address your concerns and support a reassessment of the paper’s contributions. We thank you for your constructive feedback and time!

---

> > ### Author Rebuttal · Reviewer_aBB3 · 2026-04-03
> >
> > I'd like to thank the author(s) for their thorough response.
> >
> > Overall, my concerns mostly seem to be addressed, though given the extent of the proposed changes (including a whole new background section) it is hard to judge whether they will be adequately addressed in the final version without being able to conduct another round of review.
> >
> > A minor outstanding reservation I have concerns the methodology of identifying the five categories [i]-[v]. The description provided in the rebuttal ('clustering the solution proposals identified in our analysis') is rather brief, and should be expanded on further in revisions.
> >
> > As such, I will increase my score to 4) borderline accept, with the caveat that I am unable to assess the quality of implemented revisions.

---

### Official Review · Reviewer_UemQ · 2026-03-13

**Significance:** 3
**Argument Clarity:** 2
**Rating:** 5
**Confidence:** 4

**Questions:**

line 19 "The evolution from HTTP to HTTPS demonstrates this idea: security transitioned from application-level implementation to protocol-level requirement, recognizing that inconsistent adoption creates systemic vulnerabilities". Does this evolution brings new reflections other than the one noted in its previous sentence "Protocol-level implementations shift trustworthiness from being optional features to structural requirements."

line 37. "which is the most comprehensive cross-sector framework at the time of this work". Given EU AI act the most comprehensive one, how the conclusion from this paper can be adapted to other trustworthy AI guidelines?

line 47. "Currently, these guidelines are implemented at the application layer, making their adoption inconsistent and ..." is this limitation also true in non-MCP development? Why this inconsistency become unacceptable in MCP-development?

line 83. "During deployment, organizations lack standardized risk frameworks for assessing MCP servers ..." Given this limitation is about assessing MCP servers, why i and iii is required but iv is not required.

line 98. How to define Safety in "2. Technical Robustness and Safety", given safety and security are defined separately.

Section 2 is not connected to the actions. Does this mean no actions needed or no available actions?

line 112. "Well known security issues arise across all stages of MCP lifecycle." First, it is worthy to indicate the relationship among stages. Second, given this subsection should include security issues across ALL stages, should I understand the rest subsections positions out of the MCP lifecycle?

line 132. "Gap between metadata-level trust and runtime guarantees." Is metadata-level trust the same to or different from "registry-level trust" used in the following contents.

line 133. "The threats discussed above persist due to a fundamental architectural limitation: ..." if this has been addressed, does it mean other security issues will no longer matter?

line 168. Is GDPR "General Data Protection Regulation"?

Section 9. In this action framework, is there a need to incorporate existing protocal-level or non-protocal-level actions?

**Alternative Views Section:**

Yes

**Compliance With Llm Reviewing Policy A Conservative:**

Affirmed.

**Discussion Potential:**

4

**Final Justification:**

While this paper has a number of clarity issues, the main idea is novel and significantly benefits the community.

**Paper Summary:**

This paper advocates the enforcement of trustworthy agentic AI through building a trustworthy model context protocol. Adopting the EU Ethics Guidelines for Trustworthy AI, this position identifies how trustworthiness works and critical mitigation challenges in the MCP ecosystem, and proposes nine intervention actions.

**Position:**

Yes

**Position In Title:**

Yes

**Related Work:**

3

**Strengths And Weaknesses:**

Strengths:
It targets an emerging problem.
Overall, the idea is well-conveyed, and the proposed actions are rational.

Weaknesses:
It lacks clarifications that can help distinguish the trustworthy agentic AI from trustworthy AI. Namely, the current version mainly addresses how trustworthiness works in agentic AI but fails to clarify how trustworthiness shifts or the critical challenges brought by these shifts.
Principles are discussed separately without considering cross-effect.
Lack suppliments for profound understanding on concepts and their connections.

**Support:**

3

---

> ### Author Rebuttal · Authors · 2026-03-30
>
> Dear Reviewer,
>
> thank you very much for your review. Please find our (slightly condensed due to character limits) rebuttal here:
> >> line 19[...]
>
> The HTTPS example is intended as a concrete illustration rather than a new argument. We will streamline this to prevent redundancy.
> >> line 37[...]
>
> While the EU AI Act is currently the most comprehensive cross‑sector regulatory framework, our conclusions are not limited to it. The principles we analyse, such as  transparency, robustness, data governance, and accountability, are shared across major trustworthy‑AI guidelines (e.g., OECD, NIST, ISO), and thus the MCP‑level implications we outline can be applied broadly. We will clarify this in the final verison.
> >> line 47[...]
>
> This limitation also appears in some non‑MCP developments, where trustworthiness is left to the application layer and adoption is inconsistent. However, protocols such as HTTPS, S/MIME, and HSTS show that security and privacy measures can be embedded at the protocol level to ensure consistent adoption. MCP offers a similar opportunity: by centralizing key stages of the model lifecycle, it allows trustworthiness to be implemented by design. As MCP moves toward standardization, this becomes both feasible and timely. Finally, because agentic AI will likely run on top of MCP and become ubiquitous, a trustworthiness‑by‑design approach is essential for safe everyday integration.
> >> line 83[...]
>
> When addressing the human‑oversight challenge, we focused on two relevant measures: (i) risk‑adaptive requirements and (iii) host‑level enforcement with human‑readable disclosures. These support continuous risk management across the MCP lifecycle. In contrast, (iv) targets registry‑level governance, which concerns discovery and listing rather than deployment‑time assessment. While (iv) can be seen as complementary to (i), it does not directly address the limitation discussed in the text. We will clarify this distinction in the revised version or integrate (iv) with a brief explanation.
> >> line 98[...]
>
> In the paper, we are defining robustness, safety and security as in line 99. If this distinction remains unclear, we would appreciate guidance on which aspects require further clarification.
> >> Section 2[...]
>
> We appreciate this point. We included robustness and safety for completeness, but as well‑explored areas, we only summarize existing work rather than propose new actions. We agree this rationale should be made explicit at the start of the paragraph and will do so in the revision.
> >> line 112[...]
>
> We meant to highlight the scope of existing security work, but agree the wording may imply that other issues lie outside the MCP protocol lifecycle. We will clarify this in the revision.
> >> line 132[...]
>
> Registry‑level trust comes from being listed and validated by a registry, while metadata‑level trust refers to the trust inferred from the server’s declared metadata (because metadata is usually part of establishing registry-level trust, those concepts are indeed intertwined). This paragraph aims to explain
> - what registries validate (metadata, namespace ownership, schemas)
> - what they do not validate (runtime behaviour, cryptographic integrity, execution safety)
> - why registered servers can still execute unsafe actions
> - how the MCP protocol lacks protocol‑level enforcement
> - why this creates a systemic trust gap
> But we very much agree that using both terms creates confusion, and we will clarify the formulation in the revised version.
> >> line 133[...]
>
> Fixing that gap would solve a majority of vulnerabilities discussed in the paragraph above and in existing literature - namely, cases where properly registered servers still behave unsafely at runtime. The more precise formulation that we will use in the final version would be: “A key reason these threats persist is the architectural gap between registry-level trust and runtime execution guarantees.”.
> >> line 168[...]
>
> Yes, thanks for notifying!
> >> Section 9[...]
>
> Section 9 primarily proposes MCP‑protocol‑level actions, but several of these are intentionally grounded in existing ideas, standards, or frameworks. Whenever our proposed actions build upon existing work, we reference these sources directly in the corresponding subsections. For example, our oversight strategy draws on Anthropic’s MCP elicitation primitive (line 072), and our proposal for Server Cards builds on established concepts such as Model Cards and Datasheets (lines 228 and 288).
>
> This review was particularly helpful in pointing out minor inaccuracies that can arise in deep, detail‑heavy work. We sincerely appreciate this level of attention. As MCPs increasingly form the backbone of agentic AI systems, we see this topic as timely(or urgent). We will incorporate your suggestions in the final version and hope these revisions address your concerns and support a reassessment of the paper’s contributions. Thank you for your constructive feedback and time!

---

> > ### Author Rebuttal · Reviewer_UemQ · 2026-04-05
> >
> > > lilne 98: follow-up
> >
> > Therefore, Section 2 discusses technical robustness and safety only, but should not include technical security. Given this, how to understand line 112?
> >
> > > missed weaknesses:
> >
> > * It lacks clarifications that can help distinguish the trustworthy agentic AI from trustworthy AI. Namely, the current version mainly addresses how trustworthiness works in agentic AI but fails to clarify how trustworthiness shifts or the critical challenges brought by these shifts.
> >
> > * Principles are discussed separately without considering cross-effect. Lack suppliments for profound understanding on concepts and their connections.

---

### Official Review · Reviewer_tTfk · 2026-03-13

**Significance:** 3
**Argument Clarity:** 4
**Rating:** 4
**Confidence:** 2

**Questions:**

Please refer to my weakness section.

And in addition, a lingering question is - the authors propose that the system could be make opt-in. Will that result in perverse incentives? For example, let's consider the `costPriority`- can this be gamed? a) be gamed by advertising fake numbers and NOT following up with an audit? How would this be penalized? There wasn't an argument on what happens if you refuse an audit? b) would this lead to incentives for ALL MCP providers to present a value or for NO MCP providers to present a value? c) Will this system break down if there are `K` bad actors causing a 51% attack since spinning up MCP servers (duplicate functionality, slightly different name) is quite cheap?

What happens when : the MCP is an OSS repository which is archived - i.e. can be used but is NO longer in active maintenance? And then, say, concurrently the regulation changes? Will it deem the MCP service to be out of favor?

**Alternative Views Section:**

Yes

**Compliance With Llm Reviewing Policy A Conservative:**

Affirmed.

**Discussion Potential:**

3

**Paper Summary:**

The authors analyze MCP through the EU Commission’s Ethics guidelines for trustworthy AI, and identify:  shifts in how trustworthiness works, challenges these shifts create, and recommend 5 intervention points for protocol-level mechanisms to aid in ecosystem adoption. Although it is NOT clearly stated in the title (as necessary), the position as clarified in the abstract is : building trustworthy MCP enables
responsible agentic AI deployments.

**Position:**

Yes

**Position In Title:**

No

**Related Work:**

3

**Strengths And Weaknesses:**

Strengths
+ The paper's core insight is that MCP presents itself as a natural interface point for trustworthy AI as it is a standardized interface between the LLM and the tools.
+ Going beyond the issues, the paper presents 5 concrete intervention points graded by scope, while proactively acknowledging in alternate views that this could be cumbersome, and hence countering that with making this opt-in.

Weakness
- The authors premise is that "MCP reduces human-in-the-loop oversight by design, which is not necessarily a violation of the EU Com-
mission’s guidelines." lines 074-076. I am not sure I agree with this premise. For such a sweeping statement, I would expect the authors to make a formal argument, or at least demonstrate that this is the case.
- The first central concern of "Technical oversight mechanisms fail to support informed judgment" seems to cherry-pick, what is the protocol, from what is "not best practises". For example, the authors argue that "..this requirement is not enforced at the protocol’s transport layer and is instead delegated to implementation, ... In practice, oversight is weakened by persistent informational asymmetries: requests are expressed in protocol-level terminology, .." I would argue that the abstraction / interface of implementation cannot be the root case for claiming something is "not enforced". For example, in networking, you could enforce firewalls at the application layer (at the egress), or in switch application layers (e.g., in Intel Tofino's), or as a mask in TCP. All of these are valid ways to do so, and the fact that users often "mess up" the implementation in one or the other isn't a fair argument on what the protocol can or cannot support. It is valid argument on what allows for great user experience - which is already covered in the next topic of Human intent can degrade across translation steps.
- I find some of the core arguments to be in-conflict with each other. For example, the authors claim that "..where community-driven updates vary in quality and timing.." and also the "MCP is a natural interface to reduce environmental costs" requires "MCP registries
could optionally publish publisher-provided cost and energy metadata for servers, while hosts log per-workflow usage estimates to enable auditing" which means, it's essentially self-audited? Or who would the auditing agencies be, and what would the incentives be for agencies to audit an essentially "community driven" effort?

**Support:**

4

---

> ### Author Rebuttal · Authors · 2026-03-30
>
> Dear Reviewer,
>
> thank you very much for your review. Please find our rebuttal here:
>
> >> The authors analyze MCP through the EU Commission’s Ethics guidelines for trustworthy AI, [...] Although it is NOT clearly stated in the title [...].
>
> We appreciate the reviewer’s observation. We chose the title “Let’s build a Trustworthy Model Context Protocol”, because it is in line with our Call to Action. However, it doesn’t fully convey the position of the paper. Therefore, we will change the title of the revised version into “Trustworthy MCP Enables Responsible Agentic AI”
>
> >> The authors premise is that "MCP reduces human-in-the-loop oversight by design, [...]  I am not sure I agree with this premise. [...].
>
> Thank you for raising this point. Our intention was not to claim that reduced human‑in‑the‑loop oversight is inherently compliant, but rather that the EU Commission’s guidelines allow for risk‑based and context‑dependent oversight mechanisms. Those mechanisms could be risk‑adaptive approval workflows [i], standardized disclosures of intent and scope [ii], and host‑enforced oversight rules that preserve meaningful human control throughout execution [iii]. We argue, supported by literature from law and related disciplines, that the main concern is not the reduced human-in-the-loop, but whether such a reduction results in misalignment of human intent and agency.
> We agree that this reasoning should be articulated more clearly and formally. In the revised version, we will clarify the premise and provide a short justification summarizing how the EU guidelines distinguish between 1) meaningful human oversight and 2) continuous, manual human intervention. This will make it clear that MCP’s reduction of human‑in‑the‑loop actions does not automatically constitute a violation, provided that appropriate mechanisms are in place.
>
> >>  The first central concern of "Technical oversight mechanisms fail to support informed judgment" seems to cherry-pick, what is the protocol, from what is "not best practises". [...]
>
> We agree that our phrasing in the subsection “Technical oversight mechanisms fail to support informed judgment” may suggest a protocol‑layer criticism that is stronger than intended. You are right that enforcement can reasonably occur at different layers.
> Our point is narrower: MCP leaves the semantics and granularity of consent and intent visibility entirely to host implementations, resulting in substantial variation in how much informed oversight users receive. We are not critiquing host‑side enforcement; rather, the absence of protocol‑level affordances means oversight can differ widely even among compliant hosts.
> In light of this comment, we will soften and clarify the subsection. Specifically, we will:
> - adjust the title to focus on how MCP shifts where oversight is exercised, and whether meaningful human agency is preserved (maybe “MCP and the challenge of informed oversight”)
> - remove the transport‑layer sentence, since it distracts from our intended point;
> - clarify that the core issue is variability in how hosts present consent and intent, not that enforcement occurs at the host layer;
> - emphasize that standardized, machine‑readable metadata could support more consistent, meaningful oversight across hosts.
>
> >> I find some of the core arguments to be in-conflict with each other. [...]
>
> We agree that our current Eco Mode suggestion under-specifies who would audit relevant disclosures in a community-driven ecosystem. Our intended claim is narrower than the current wording suggests: MCP does not itself solve environmental accountability, but it can provide a useful interface for standardizing the disclosure and logging mechanisms through which accountability can be implemented. We will clear this up in the revision.
>
> >> And in addition, a lingering question is - the authors propose that the system could be make opt-in. Will that result in perverse incentives? [...]
>
> We agree that the current Eco Mode suggestion under-specifies the relevant incentive and enforcement model. As noted above, our claim is not that Eco Mode solves environmental well-being concerns on its own, but rather that MCP provides a useful interface for standardizing accountability hooks. In the revised version, we will clarify that such signals should be treated as claims rather than facts. While Eco Mode does not eliminate gaming, it creates a shared exposure surface through which external governance mechanisms can assess claims, attach consequences to misleading or unsupported disclosures, and, where appropriate, support legal accountability.
>
> Given that MCPs are increasingly becoming the backbone of emerging agentic AI systems, we see this topic as both timely (or even urgent) and highly relevant. We will incorporate your suggestions into the final version and hope this and our clarifications address your concerns and support a reassessment of the paper’s contributions. We thank you for your constructive feedback and time!

---

> > ### Author Rebuttal · Reviewer_tTfk · 2026-04-06
> >
> > I thank the authors for their rebuttal.
> >
> > >"such signals should be treated as claims rather than facts."
> >
> > Given that the incentive structure is to `boost the claims` so as to speak, and given a lack of enforcement, it would be worth a discussion point on how to best leverage such a system.
> >
> > > "it creates a shared exposure surface"
> >
> > Yup - agree!

---

### Decision · Program_Chairs · 2026-04-30

**Decision:**

Accept (regular)

**Comment:**

Reviewers all appreciated the importance and timeliness of the problem addressed by this position paper: ensuring trustworthy MCP. The main issue identified was that the paper seems to try to cover too much ground and as a result is difficult to read and its claims come off as a but vague and unfocused. However, the reviewers felt that these were not serious enough flaws to warrant rejection. The importance and relative neglectedness of the topic make up for the other deficiencies.